



# Aerosol impacts on warm-cloud microphysics and drizzle in a moderately polluted environment

Ying-Chieh Chen[1], Sheng-Hsiang Wang[1,2,*], Qilong Min[3], Sarah Lu[3,4], Pay-Liam Lin[1], Neng-Huei Lin[1,2], Kao-Shan Chung[1], Everette Joseph[3,5]

[1]Department of Atmospheric Sciences, National Central University, Taoyuan, Taiwan
[2]Center for Environmental Monitoring and Technology, National Central University, Taoyuan, Taiwan
[3]Atmospheric Sciences Research Center, State University of New York, University at Albany, Albany, NY
[4]Also at Joint Center for Satellite Data Assimilation, Boulder, CO
[5]Now at National Center for Atmospheric Research, Boulder, CO

*Correspondence to*: Sheng-Hsiang Wang (carlo@g.ncu.edu.tw)

**Abstract.** Climate is critically affected by aerosols, which can alter cloud lifecycles and precipitation distribution through radiative and microphysical effects. In this study, aerosol and cloud properties datasets from MODIS onboard Aqua satellite and surface observations, including aerosol concentrations, raindrop size distribution, and meteorological parameters, were used to statistically quantify the effects of aerosols on low-level warm cloud microphysics and drizzle over northern Taiwan during fall seasons (from October 15 to November 30 of 2005–2017). Results indicated that clouds in northwestern Taiwan, which with active human activity is dominated by low-level clouds (e.g. warm, thin, and broken clouds). The observed effects of aerosols on warm clouds indicated aerosol indirect effects; increasing aerosol loading caused a decrease in cloud effective radius (CER), an increase in cloud optical thickness, an increase in cloud fraction, and a decrease in cloud top temperature under a fixed cloud water path. A quantitative value of aerosol–cloud interactions ($ACI = \partial \ln CER / \partial \ln \alpha$, changes in CER depend on changes in aerosols) were calculated to be 0.07 for our research domain. ACI values varied between 0.09 and 0.06 in surrounding clean and heavily polluted areas, respectively, which indicated that aerosol indirect effects were more sensitive in the clean area. Analysis of raindrop size distribution observations during high aerosol loading resulted in a decreased frequency of drizzle events, redistributed cloud water to more numerous and smaller droplets, and reduced collision-coalescence rates. However, in the scenario of light precipitation ($\leq 1$ mm h$^{-1}$), high aerosol concentrations drive raindrops towards smaller droplet sizes and increase the appearance of drizzle drops. This study used long-term surface and satellite data to determine aerosol variations in northern Taiwan, effects on the clouds and precipitations, and applications to observational strategy planning for future research on aerosol–cloud–precipitation interactions.

## 1 Introduction

Since the industrial revolution, the quantity of aerosols produced by human activities has increased significantly. Aerosols are most concentrated in areas with frequent industrial activities and high biomass burning because of the short lifetime of aerosols (Textor et al., 2006). The effect of aerosols on climate is recognized as significant (Charlson et al., 1992; Kiehl and





Briegleb, 1993; Penner et al., 2001; Ramanathan et al., 2001; Ramaswamy et al., 2001). Aerosols can alter cloud properties and the subsequent adjustments, also known as the aerosol indirect effect (Warner and Twomey, 1967; Twomey, 1974; Albrecht, 1989; Lohmann and Feichter, 2005). The responses of convective and boundary layer clouds contribute to the spread of global cloud feedbacks in general circulation models (GCMs), with a dominant role of intermodel differences in the response

of low-level clouds (Bony et al., 2006). The concentration of aerosol particles and cloud condensation nuclei (CCN) provides a valuable link between aerosol and cloud. Aerosols can alter warm cloud characteristics through radiative and microphysical effects, which has a substantial effect on climate. However, studies have demonstrated that the global model significantly overestimates the frequency of drizzle Stephens et al. (2010), which brings into question the accuracy of aerosol–cloud interactions (ACI)s in models. Therefore, observational studies of aerosol and cloud microphysical properties are crucial for

clarifying the relationship between aerosols and the microphysical process of clouds and evaluating the accuracy of model simulations.

Jones et al. (2009) emphasized that ACI should be explored at the regional scale, because the aerosol type, concentration, and meteorological conditions differ depending on the area. Numerous studies have used the aerosol concentration and cloud droplet size to investigate ACIs at global or regional scales. A negative correlation between aerosols and cloud drop size has

been observed in global (Bréon et al., 2002; Myhre et al., 2007; Nakajima et al., 2001) and regional scale (Costantino and Bréon, 2010; Ou et al., 2012) studies. Sekiguchi et al. (2003) and Grandey and Stier (2010) have used global satellite data and identified different correlations (positive, negative, or weak) between aerosol optical depth (AOD) and cloud effective radius (CER) depending on the location of the observation.

The effect of aerosols on cloud microphysics can be observed using satellite data (Krüger and Graßl, 2002; Menon et al.,

2008; Rosenfeld et al., 2014; Saponaro et al., 2017; Sporre et al., 2014). Using satellite-based precipitation observations from the Tropical Rainfall Measuring Mission (TRMM), Rosenfeld (1999) demonstrated that aerosols derived from biomass burning suppress warm rain processes. Ground observations and models have revealed a strong correlation between CER and AOD (Feingold et al., 2001; Grandey and Stier, 2010; Yuan et al., 2008). Aircraft observations over the Amazon basin demonstrated decreased in-cloud droplet sizes and a delay in precipitation onset when a large quantity of aerosols entered the cloud (Andreae

et al., 2004). The effects of aerosols in suppressing drizzle have been identified in field experiments on stratocumulus clouds over the northeastern Atlantic Ocean (Albrecht et al., 1995; Wood, 2005), northeastern Pacific Ocean (Lu et al., 2007; Lu et al., 2009; Stevens et al., 2003; VanZanten et al., 2005), and southeastern Pacific Ocean (Bretherton et al., 2010; Comstock et al., 2004; Wood et al., 2011). Moreover, model simulations revealed that polluted environments could suppress drizzle in warm clouds (Ackerman et al., 2004; Guo et al., 2011; Wang et al., 2011a; Wang et al., 2011b). Although numerous studies

have used observations and model simulations to discuss the indirect effects of aerosols, the interaction mechanism between aerosols and clouds remains unconstrained in the global climate system.

Huang et al. (2007) used a regional coupled climate–chemical–aerosol model for the East Asia region and determined that the aerosol indirect effect significantly reduced precipitation in autumn and winter. Menon et al. (2002) used a global climate model to study the effects of aerosols in China and India and reported that anthropogenic aerosols increase precipitation





in southeastern China but inhibits precipitation in northeastern China. Furthermore, Giorgi et al. (2003) used a coupled regional chemistry–climate model to assess the direct and indirect effects of anthropogenic sulphates on East Asian climates. Results indicated that aerosol indirect effects were largely dominant in inhibiting precipitation. Takemura et al. (2005) used a global aerosol transport-radiation model coupled to a general circulation model and determined that the indirect effect had a strong

signal in regions with large quantities of anthropogenic aerosols and cloud water.

These studies have demonstrated a significant correlation between aerosols and cloud microphysics. They further demonstrated an indirect effect of aerosols on regional precipitation. However, the aerosol type, concentration, and characteristics vary by region. Taiwan is an island with a high population density, a complicated topography, a climate that ranges from tropical in the south to subtropical in the north. These characteristics result in substantially complex microphysical

processes between aerosols and clouds. In this study, we aimed to systematically analyze aerosols, cloud optical properties, and precipitation characteristics by integrating satellite and surface observation data over northern Taiwan to investigate the following questions: (1) How do aerosols affect cloud microphysical properties in response to different pollution conditions? and (2) How do aerosols affect the frequency of drizzle and the change in precipitation distribution? In Sect. 2, we describe data and methodology. In Sect. 3, we present results and discussion. Findings are summarized in Sect. 4.

## 2 Data and methodology

### 2.1 Study area and time period

Our study domain, northern Taiwan, covers the area 24.5°–25.8° N and 120.8°–122.2° E (Fig. 1) with a population of approximately 10 million. The emissions of this area are considered a combination of urban and industrial activities. For this area, air quality decreases in fall when precipitation is less and air masses become more stagnant. Moreover, the results of

Huang et al. (2007) suggested that aerosol indirect effects frequently happened in fall. Therefore, we chose the data period from 15 October to 30 November between 2005 and 2017 (611 days in total) to explore aerosol effects on cloud microphysics and drizzle. To prevent the effect of typhoons on the analysis, typhoon alarm days (21–23 October 2010, Typhoon Megi) issued by the Central Weather Bureau were excluded in this study.

### 2.2 Surface measurement data

Hourly meteorological (i.e. temperature, relative humidity, rainfall, wind direction, and wind speed) and $PM_{2.5}$ concentration data collected from Taiwan EPA Pingzhen site (24.95° N, 121.20° E) and one-minute raindrop size distribution Joss–Waldvogel Disdrometer (JWD) data obtained from National Central University (NCU) (24.968° N, 121.185° E) observatory were used. The NCU and EPA Pingzhen sites are located near each other at the centre of the study domain. The $PM_{2.5}$ concentration was measured using the MetOne BAM-1020 Beta Attenuation Monitor. The JWD measures the number of rain

droplets every minute by using 20 bin sizes of 0.359–5.373 mm (n1–n20: 0.359, 0.455, 0.551, 0.656, 0.771, 0.913, 1.116,





1.331, 1.506, 1.665, 1.912, 2.259, 2.584, 2.869, 3.198, 3.544, 3.916, 4.350, 4.859, and 5.373 mm). To ensure data quality, observations were discarded when the rain rate was lower than 0.1 mm h$^{-1}$ (Greenberg, 2001; Seela et al., 2017).

## 2.3 Satellite data

Cloud and aerosol data from NASA Aqua satellite, moderate-resolution imaging spectroradiometer (MODIS) collection 6 level 2 products (MYD06 for clouds and MYD04 for aerosols) were used in this study. Data were downloaded from https://modis.gsfc.nasa.gov/data/. Data on cloud properties included cloud optical thickness (COT), CER, and cloud water path (CWP), all of which had a resolution of 1 km, as well as cloud fraction (CF), cloud-top pressure (CTP), cloud-top temperature (CTT), and cloud phase infrared (CPI), all of which had a resolution of 5 km. CWP included liquid water path and ice water path (CWP = LWP + IWP). For aerosol data, AOD with a resolution of 10 km was used. Descriptions of parameters and products are presented in Table 1. To ensure spatial resolution consistency between data sets, data were interpolated to a coarse resolution of 0.1° × 0.1°.

## 2.4 Data screening and grouping

Satellite aerosol data were not retrieved when conditions were overcast, except when aerosols were above clouds. To compensate for this limitation, densely available surface PM$_{2.5}$ data in the study domain was used. The spatial homogeneity of the PM$_{2.5}$ concentration was examined based on the correlation between the Pingzhen site with the 30 air quality monitoring sites in the north part of Taiwan. Results indicated that correlation coefficients were higher than 0.6 and 0.8 for northern Taiwan and the research area (24.6°–25.2° N and 120.9°–121.5° E), respectively, indicating that PM$_{2.5}$ data from the Pingzhen site accurately represented aerosol distribution over our research domain (Fig. 1).

Clouds and their microphysics properties in the afternoon may be affected by aerosols in the morning to noontime. PM$_{2.5}$ data between 10:00 and 14:00 were averaged as a measure of daily PM$_{2.5}$ concentrations to accord with Aqua satellite data (overpass time is approximately 13:30 local time). Furthermore, the 20$^{th}$ percentile of daily average PM$_{2.5}$ data was defined as clean days, and the corresponding PM$_{2.5}$ concentration was 11.2 μg m$^{-3}$ for 123 days. The 80$^{th}$ percentile of daily average PM$_{2.5}$ data was defined as polluted days, and the corresponding PM$_{2.5}$ concentration was 34.6 μg m$^{-3}$ for 121 days. Polluted days were further divided into three groups: slightly polluted (40 days), moderately polluted (40 days), and heavily polluted (41 days) with PM$_{2.5}$ concentrations of 34.6–39.9, 39.9–52.3, and 52.3–110 μg m$^{-3}$, respectively.

A study (Wang et al., 2010) reported that the vertical aerosol distribution for the study region in autumn was mainly within 2 km. For ACI at a local scale, clouds that occurred at the same level were targeted. Therefore, only clouds with CTP ≥ 800 hPa and CPI = 1 (water cloud) were included, thereby ensuring that only warm clouds were analyzed.

To quantify ACI, the commonly used formula proposed by Feingold et al. (2001) was employed, as illustrated in Eq. (1). This equation calculates how a change in aerosols affects CER at a constant CWP.



$$\text{ACI} = -\frac{\partial \ln \text{CER}}{\partial \ln \alpha}\big|_{CWP}, \tag{1}$$

where α represents the proxy for the quantity of aerosols, using either $PM_{2.5}$ or AOD values. Positive ACI values indicate that the effect of a change in CER depends on increased aerosols, and vice versa. The ACI value approaching 0 indicates that the relationship between CER and aerosol indirect effects is not significant. The ACI calculation should be under the fixed range

of CWP in Eq. (1). Therefore, the CWP population density distribution was divided into ten groups (Fig. 2), with each group representing 10 % of CWP data.

Data on the precipitation raindrop size distribution obtained from JWD were further processed. The daily rainfall amount was defined as the sum of precipitation from 10 am to the next day at 10 am to investigate the consequential process of aerosol-cloud-precipitation in a day scenario by using currently available data sets. The AMS Glossary (Huschke, 1959) defines drizzle

as very small, numerous, and uniformly dispersed water drops that may appear to float in currents. In contrast to fog droplets, drizzle falls to the ground. In weather observations, drizzle is classified as (a) "very light," comprised of scattered drops that do not entirely wet an exposed surface regardless of the duration; (b) "light," the rate of fall being traced to 0.25 mm $h^{-1}$; (c) "moderate," the rate of fall being 0.25–0.50 mm $h^{-1}$; and (d) "heavy," the rate of fall exceeding 0.5 mm $h^{-1}$. When the precipitation equals or exceeds 1 mm $h^{-1}$, all or part of the precipitation is considered rain. The threshold for rain intensity was

set at 1 mm $h^{-1}$ to focus on the effect of aerosols on drizzle. Drizzle drops are conventionally 0.5 mm or less in diameter; therefore, JWD data in n1 (0.359 mm) and n2 (0.455 mm) channels were summarized as drizzle precipitation.

## 3 Results and discussion

### 3.1 Overall aerosol, cloud, and meteorological characteristics

To explore the effect of aerosols on cloud microphysics and the subsequent precipitation, a general understanding of aerosol

quantities, cloud microphysics, and precipitation characteristics over the study region is crucial. Figure 3 illustrates the spatial distribution of mean aerosol and cloud parameter values (including AOD, COT, CWP, CF, CER, and CTP) over northern Taiwan from October 15 to November 30, 2005–2017. The mean AOD reached 0.6 in northwest Taiwan because of the high density of human activities, whereas lower AOD values (less than 0.2) were observed over the Xueshan Range (the green triangle in Fig. 3a).

The characteristics of clouds are affected by the prevailing northeast wind and topography; therefore, clouds generally have higher top heights and more significant coverage over northeastern Taiwan compared with northwestern Taiwan. The mean COT, CWP, and CER in our study area had ranges of 10 g $m^{-2}$, 60–120 g $m^{-2}$, and 13–14.5 μm, respectively. The mean CF was 0.6–0.7. Most of the CTP was higher than 850 hPa, suggesting low-level clouds (e.g., warm, thin, and broken clouds). For the spatial distribution of data availability, larger quantities were collected in our main research domain, indicating robust

statistical results.



The characteristics of the surface $PM_{2.5}$ concentration and meteorological parameters for clean and polluted days were also analyzed. We collected 1189 hours of rainfall data, and the number of other meteorological parameters data was approximately 14,000. The mean values of temperature, relative humidity, rainfall, wind speed, and $PM_{2.5}$ concentrations were 22.3 °C, 74.9 %, 1.4 mm h$^{-1}$, 3.2 m s$^{-1}$, and 23.4 μg m$^{-3}$, respectively (illustrated in Fig. 4). The prevailing wind direction was

northeast. During clean days, the aforementioned mean values were 22.2 °C, 79.3 %, 1.5 mm h$^{-1}$, 3.6 m s$^{-1}$, and 9.9 μg m$^{-3}$, respectively, compared with the mean values of 22.5 °C, 72.5 %, 1.4 mm h$^{-1}$, 2.7 m s$^{-1}$, and 43.3 μg m$^{-3}$, respectively, on polluted days. Overall, compared with clean days, meteorological conditions on polluted days had lower relative humidity, less rainfall, more wind direction in addition to the northeast wind, and lower wind speed. However, differences were not observed in mean rainfall rates between clean and polluted days. The number of rainfall hours differed significantly with 384

hours during clean days and 115 hours during polluted days. A weaker and more disorderly direction of the wind was observed on polluted days, which suggests that pollution may be associated with more stagnant conditions.

CWP is a constraint factor for the ACI index calculation illustrated in Eq. (1). We further examined CWP variability in response to main meteorological parameters (temperature, relative humidity, and rainfall) and $PM_{2.5}$ concentrations from the Pingzhen site and CER from MODIS. We calculated the daily mean value of CWP and CER by averaging grids over the main

research area (24.6°–25.2° N, 120.9°–121.5° E). Daily meteorological parameters and $PM_{2.5}$ concentration data, described in Sect. 2.2, were used. Figure 5 illustrates the means and standard deviations of $PM_{2.5}$ and CER in 10 CWP groups. As CWP increased, the average temperature and relative humidity gradually decreased and increased, respectively. No significant correlation was identified between rainfall and CWP. The complicated relationship between $PM_{2.5}$ and CWP is illustrated in Fig. 5. $PM_{2.5}$ increased with an increase in CWP to 50 g m$^{-2}$ and then decreased, whereas CER increased at first before

decreasing and then increasing again. CWP standard deviation in group 9 ($150 \leq CWP < 297$) was smaller than in other groups, indicating that group 9 was a more stable community; thus, the subsequent analysis focused on group 9 to reduce uncertainties caused by the variability of environmental conditions and improve our understanding.

## 3.2 Aerosol effect on warm cloud properties

The effects of aerosols on warm cloud microphysics in different CWP groups for the main research domain were studied using

the ACI index (Eq. (1)). Figure 6 illustrates the ACI values and correlation coefficient (r(ACI)) of the $PM_{2.5}$ mass concentration and CER under different CWP groups. ACI was 0.07 in CWP group 9 ($150 \leq CWP < 297$) with the lowest root mean square error (RMSE = 0.23) compared with other groups. The correlation coefficient between $PM_{2.5}$ and CER in group 9 was −0.19. Positive ACI values were observed when CWP was higher than 125 g m$^{-2}$, and a higher value of ACI is associated with higher CWP groups. The negative correlation indicates an aerosol indirect effect (i.e. an increase in aerosols cause cloud droplet

radiuses to become smaller under a fixed water content). Negative ACI values are illustrated in low CWP groups (i.e. groups 1–7), which may be caused by the large deviation of CER data in smaller CWP groups. However, the low water content may reduce the effects of aerosols on warm cloud microphysics. We compared our results with the literature. Feingold et al. (2003) analyzed ACIs by using ground-based remote sensors in Oklahoma, United States, focusing on ice-free, single-layered,





nonprecipitating, and airborne insect–free clouds. Their results indicated that under the same LWP, the ACI values of seven cases were 0.02–0.16. Kim et al. (2008) conducted a three-year experiment by using ground-based remote sensors to investigate the aerosol indirect effect. Their results suggested that the ACI values of continental stratus clouds ranged from 0.04 to 0.17 in north-central Oklahoma. McComiskey et al. (2009) observed the ACI values of coastal stratiform clouds between 0.04 and
0.15 by using ground-based remote sensing data from the Atmospheric Radiation Measurement (ARM) program at Pt. Reyes, California, United States. Their findings indicated that values for the anthropogenic polluted area in the current study were on the lower end but within a reasonable range, despite an ACI value of 0.07 in the CWP group 9 ($150 \leq CWP < 297$).

Because of the distinct ACI signal at CWP group 9, we further explored the effect of aerosols on cloud microphysical parameters. The difference in cloud microphysics parameters between polluted days and clean days over the main research
area (24.6°–25.2° N, 120.9°–121.5° E) was calculated. Compared with clean days, the COT, CER, CF, and CTT exhibited changes of +9.53, −2.77 μm, +0.07, and −1.28 K on polluted days (Fig. 7). These findings indicate that higher $PM_{2.5}$ concentrations may cause smaller cloud droplet sizes under a high CWP environment, which accords with the concept of the aerosol indirect effect. Consequently, smaller droplets will reduce collision–coalescence rates and suppress precipitation, thereby increasing the cloud lifetime, fraction, and optical depth.

The relationship between the cloud vertical profile and aerosols was also studied. Figure 8 displays CWP group 9 ($150 \leq$ CWP < 297) results, the occurrence frequency (%) as a function of CTT and CER, and the vertical profiles of CER occurrence frequency on clean and polluted days. On clean days, the highest occurrence frequency was located at a CER value of approximately 8 μm, which was similar on polluted days. However, CTT was lower on clean days, and the CER occurrence frequency increased overall. In other words, when the CTT was lower, the occurrence frequency of larger CER was higher.
This phenomenon could be caused by the onset of water cloud generation when updrafts dominate, causing adiabatic growth, and the reduction of in-cloud depth droplet size increases (Saito et al., 2019). On polluted days, the CER barely changed with a reduction in CTT. CTT between 285 and 288 K exhibited a higher occurrence frequency during polluted days, whereas clean days had a higher frequency of CTT between 282 and 285 K. These results suggest high aerosol concentrations introduced higher concentrations of CCN, produced more liquid particles at warmer CTT, and inhibited the development of cloud droplet
size.

### 3.3 Effect of different polluted conditions on ACI

We further explored the effect of aerosols on cloud microphysics under different polluted conditions. We investigated ACI from two perspectives, considering different polluted levels and considering different polluted areas. First, we divided polluted days into three equal groups: slightly, moderately, and heavily polluted days. We then calculated ACI values by using RMSE
and correlation coefficients (denoted with r(ACI)) of $PM_{2.5}$ and CER under different CWP groups and at different polluted levels for the main research domain. As illustrated in Fig. 9a, three polluted levels exhibited similar trends, but stronger ACI signals (larger ACI slope and absolute r(ACI) values) were observed for heavily polluted cases compared with moderately and slightly polluted days. On heavily polluted days (red line), when the CWP was higher than 50 g m$^{-2}$, the ACI value increased



as CWP increases. When CWP increased to group 8, the ACI value was positive for polluted days, whereas ACI values for slightly and moderately polluted days continued to increase in groups 7 to 9 but decreased in group 10. For CWP in groups 7–10, the ACI values of heavily polluted days were higher than the ACI value of slightly and moderately polluted days, especially in group 10. Notably, CWP at group 9 ($150 \leq$ CWP $< 297$) displayed consensus on ACI and r(ACI) values, indicating that

clouds with a CWP range of 150–297 g m$^{-2}$ were sensitive to aerosol indirect effects. Under high pollution, aerosols had a larger effect on cloud microphysics and larger positive ACI values (0.08, 0.07, and 0.06 for heavily, moderately, and slightly, respectively).

The effects of aerosols on cloud microphysics over the land and ocean (denoted with magenta and blue square boxes, respectively, in Fig. 3) are discussed. Because of the lack of surface observation of PM$_{2.5}$ over the ocean, we used AOD from

MODIS/Aqua as the aerosol proxy in Eq. (1) for the ACI calculation. To ensure the reliability of calculations, we computed ACI in the primary research area (24.6°–25.2° N, 120.9°–121.5° E) based on different aerosol proxies (i.e. AOD and PM$_{2.5}$ concentration). As illustrated in Fig. 9b, in CWP groups 1–8, ACI values evaluated with AOD had larger values than those evaluated with PM$_{2.5}$; the difference was the largest in CWP group 2 (0.22). For positive ACI ranges, ACIs estimated with AOD were positive for CWP groups 7–10, whereas ACIs computed with PM$_{2.5}$ were positive after CWP group 8. In CWP

groups 8–10, differences in ACI values became smaller, especially in group 9. We focused on group 9, which had an ACI value using PM$_{2.5}$ of 0.07 and an ACI value using AOD of 0.06; the difference between the two calculations is only 0.01.

The effects of aerosols on cloud microphysics in polluted (i.e. land) and remote (i.e. ocean, mean AOD of 0.31) areas can be assessed further by using the ACI value with AOD as an aerosol proxy. We defined the main research area of 24.6°–25.2° N and 120.9°–121.5° E as the polluted area (Fig. 3a magenta box) and 25.2°–5.8° N and 120.9°–121.5° E as the remote area

(Fig. 3a blue box). As illustrated in Fig. 9c, ACI values and correlation coefficients between mean AOD and CER were calculated in remote and polluted areas. Comparing ACI values between polluted and remote areas demonstrated that ACI values were higher in the polluted area in CWP groups 1–5. In this CWP interval, the ACI values of the remote area increased with an increase in CWP, whereas the ACI values of the polluted area changed significantly. In CWP groups 6–10, the ACI values of the remote area became more pronounced than the polluted area. The positive and increasing tendency of ACI values

was observed in larger CWP groups (>7) in two areas, suggesting that the environmental condition (i.e. water vapor) was critical in aerosol indirect effects. In CWP group 9, ACI values were 0.09 and 0.06 for remote and polluted areas, respectively, indicating that aerosol indirect effects are stronger remote areas (i.e. lower aerosols). These results are consistent with a study (Saponaro et al., 2017) that reported that large aerosol concentrations can saturate the effect of ACI causing a lower ACI value.

## 3.4 Aerosol effect on precipitation

The presence of aerosols enhances the concentration of condensation nuclei under the quantitative water content, which increases the cloud droplet number, reduces CER, and increases COT and CF. These changes subsequently alter the cloud lifetime and the precipitation process. This section further explores their consequential influence on precipitation. High-time



resolution (one-minute) JWD and PM$_{2.5}$ datasets were used to investigate the effects of aerosols on the raindrop size distribution, rainfall, and cloud lifetime.

The number of sample occurrences under different raindrop size classifications for clean and polluted days (not displayed here) were significantly higher on clean days, suggesting that a cleaner environment could be favourable for raining, compared
with polluted environments. We further calculated the daily-averaged droplet number in each raindrop size classification for polluted and clean days. The difference is plotted as Fig. 10a. The results illustrate that during polluted days, the droplet numbers appear lower for the smaller raindrop bins ($\leq$ n8) and higher for the larger raindrop bins ($>$ n8). A significant reduction in droplet numbers (decreased from 68 in clean days to 56 in polluted days) was observed at the n2 bin, which represents the reduction in drizzle. Our preliminary findings suggest that cloud condensation nuclei may have competing effects on water
uptake under aerosol-laden air and cloud water content limited conditions, which would alter the precipitation processes.

To investigate the effect of the high aerosol concentration on light rain, we created a similar plot to those that we had previously constructed but only for precipitation of less than or equal to 1 mm h$^{-1}$ (Fig. 10b). Our statistics for the droplet number concentration indicated that raindrop occurrence at n1 and n2 (i.e. drizzle) accounted for over 50 % on both polluted and clean days, indicating that drizzle drops were a common raindrop type when rainfall was $\leq$1 mm h$^{-1}$. Instead of the daily
average number used in Fig. 10a, the cumulative number distribution of each raindrop size for clean and polluted days was calculated. We then normalized the data by computing the percentage of droplet numbers in each raindrop size class to the total number. The difference between polluted and clean days is illustrated in Fig. 10b. We determined that when rainfall was $\leq$1 mm h$^{-1}$, polluted days accounted for a more significant proportion of n1 and n2 than clean days (especially in the raindrop size distribution n1, which accounted for 2.35 % more), whereas polluted days accounted for a significantly lower proportion
of n3 to n8 (Fig. 10b). These results indicate that if precipitation is lower than or equal to 1 mm h$^{-1}$, a high aerosol concentration drives raindrops to move towards smaller drop sizes, which increases the appearance of drizzle drops.

A modeling study (Huang et al., 2007) revealed that the second aerosol indirect effect (a large number of small droplets are generated and enhanced aerosols reduce the cloud precipitation efficiency) significantly reduces fall and winter precipitation from 3 % to 20 % across East Asia. In this study, we used observational data (i.e. JWD) to analyze the difference
between the average daily rainfall of polluted and clean days in different CWP groups and explore whether the increase in aerosol loading inhibits precipitation. Figure 11a demonstrates that the daily rainfall difference between polluted and clean days varies largely between CWP groups 1–7, which may be because of the smaller sample number in the interval. However, the average daily rainfall on clean days consistently exhibits higher values in CWP groups 8–10 compared with polluted days. In CWP group 9 (150 $\leq$ CWP $<$ 297), the daily average rainfall on polluted days (1.4 mm) decreased by 6.8 mm compared
with clean days (8.2 mm). Our findings suggest that under the fixed cloud water content, precipitation tends to decrease in high aerosol loading environments, which echoes findings reported in Sect. 3.2.

Furthermore, we analyzed the hourly rainfall rate of CWP group 9 (150 $\leq$ CWP $<$ 297) for clean and polluted days to explore the effect of aerosol on cloud lifetime (i.e. the cloud lifetime effect (Albrecht, 1989; Pincus and Baker, 1994; Lohmann and Feichter, 2005)). Figure 11b illustrates the 24-hour rainfall rate trends for clean and polluted days. On clean days, rainfall





is randomly distributed throughout the entire day with a notably larger rainfall rate observed after 4 am, whereas no rainfall was observed during daytime on polluted days, and a relatively weak rainfall rate started early in the night. Our preliminary results suggest that aerosols can suppress and delay precipitation. Under a constant liquid water content, a higher aerosol concentration redistributes cloud water to more numerous and smaller droplets, reducing collision–coalescence rates, which

in turn suppress precipitation and delays rainfall occurrence. Our results provide evidence of aerosol indirect effects in a highly populated island in the western Pacific.

## 4 Conclusions

Numerous studies have explored aerosol-cloud-precipitation interactions in marine stratocumulus clouds based on in-situ observations, satellite observations, and models; however, few studies have investigated clouds over a dense population and

complex topography area. In this study, we integrated numerous aerosol, cloud, and precipitation data from satellite and surface observations to quantify the effects of aerosols on low-level warm cloud microphysics and precipitation over northern Taiwan. A 13-year (2005–2017) dataset with a selected time frame (October 15 to November 30) was used in this study. In contrast to previous studies that have focused on the rainfall rate, we investigated changes in raindrop size distribution as the key variable in the effect of aerosols on precipitation.

We used surface $PM_{2.5}$ mass concentration data as aerosol proxy to determine the aerosol indirect effect. Based on the $PM_{2.5}$ mass concentration level, we split the observations into clean, polluted, slightly polluted, moderately polluted, and heavily polluted groups. The analysis of aerosol effects on clouds indicated that in CWP group 9 ($150 \leq CWP < 297$), the average of COT in the main research area increased by 9.53, CER decreased by 2.77 μm, CF increased by 0.07, and CTT decreased by 1.28 K in polluted days compared with clean days. Results illustrate that increasing the aerosol loading increases

the cloud droplet number concentration and reduces the cloud droplet size under fixed water content, thereby increasing the cloud lifetime, increasing the CF, and allowing clouds to develop further; these results are consistent with the aerosol indirect effect. The vertical distribution of warm clouds in clean and polluted days indicates that higher aerosol concentrations produced more liquid particles at lower altitude and inhibited the development of the cloud droplet size under the air-polluted condition. Moreover, the effects of aerosol on cloud microphysics in polluted (i.e. land) and remote (i.e. ocean, less polluted) areas were

investigated in CWP group 9, the ACI value of the remote area was 0.09, and the polluted area was 0.06. The ACI value in the remote area was larger than in the polluted area, indicating that clouds in the remote area were more sensitive to aerosol indirect effects. Our analysis revealed that the higher aerosol concentration redistributed cloud water to more numerous and smaller droplets under a constant liquid water content, reducing collision–coalescence rates, which further suppressed the precipitation and delayed rainfall duration. By combining the observation of raindrop size distribution, we determined that the frequency of

drizzle in the polluted conditions was decreased, whereas the high aerosol concentration caused a reduction in raindrop sizes, which increased the appearance of drizzle drops in the scenario of low precipitation ($\leq 1$ mm h$^{-1}$).



Our observational result from northern Taiwan in fall show in agreement with the aerosol indirect effects. However, we did not consider the aerosol direct radiative effect and variations caused by different weather systems in the long-term statistic. Overall, this study used long-term surface and satellite data for a preliminary understanding of aerosol variations in northern Taiwan, the effects of aerosol on the environment, and the effects of aerosols on the precipitation. We suggest
that further researches on aerosol–cloud–precipitation interactions over this area need to be carried out to fully understand the processes.

*Data availability.* The satellite data from the MODIS instrument used in this study were obtained from https://ladsweb.modaps.eosdis.nasa.gov/search/. The meteorological and $PM_{2.5}$ observation data were available from Taiwan EPA at https://erdb.epa.gov.tw/FileDownload/FileDownload.aspx. JWD disdrometer data in this study were provided by the
Planetary Boundary Layer and Air Pollution Lab. of the Dept. of Atmospheric Sciences, National Central University of Taiwan (http://pblap.atm.ncu.edu.tw/weather10.asp).

*Acknowledgements.* This research was supported by the US-Taiwan PIRE Program which supported by the Ministry of Science and Technology under grants No. MOST 104-2923-M-008-003-MY5 and U.S. National Science Foundation under contracts PIRE-1545917 managed by Dr. Everette Joseph and Dr. Pay-Liam Lin. And also supported by the Ministry of Science and
Technology under grants No. MOST 108-2111-M-008-025. We thank NASA for providing the MODIS data, and Taiwan EPA for providing the air quality and meteorological data.

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





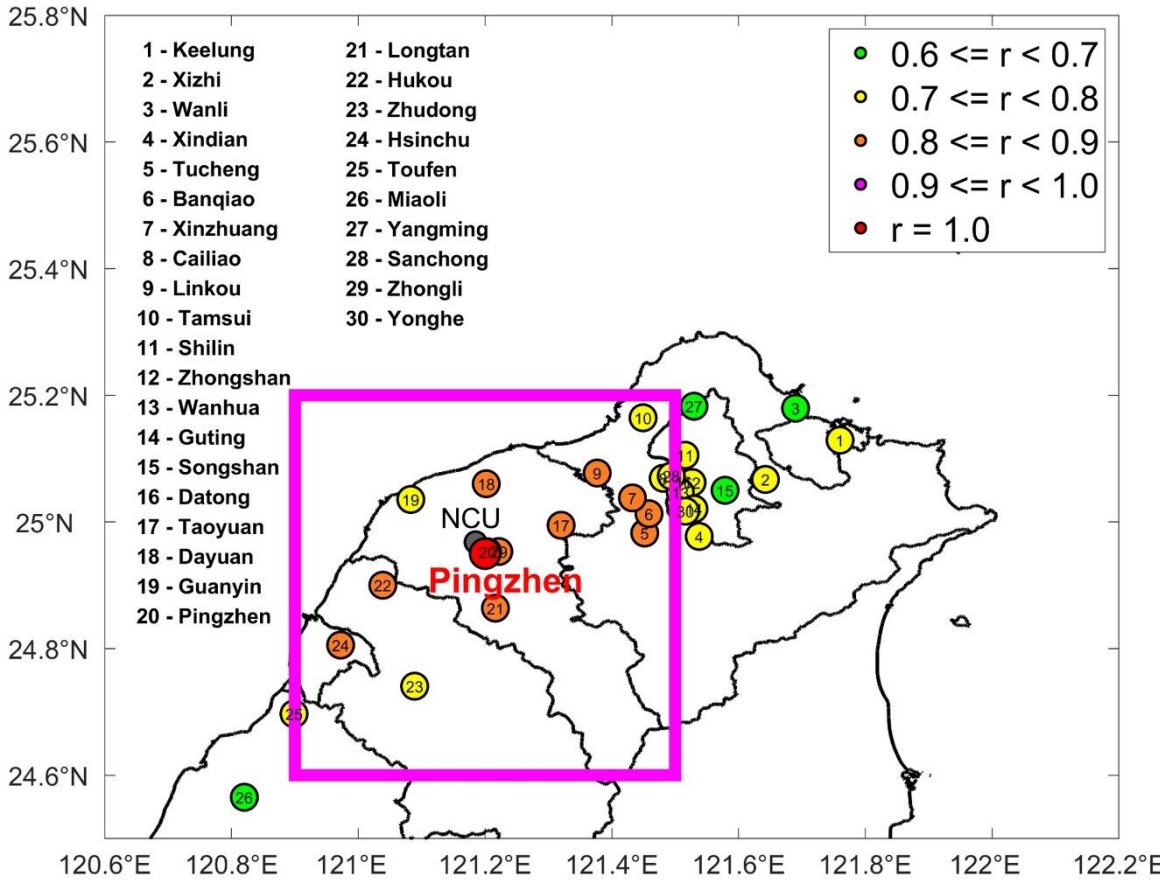

**Figure 1: Spatial correlation coefficient of the PM₂.₅ concentration between Pingzhen station and other stations. The main research area (24.6°–25.2° N, 120.9°–121.5° E) is indicated with a magenta box.**





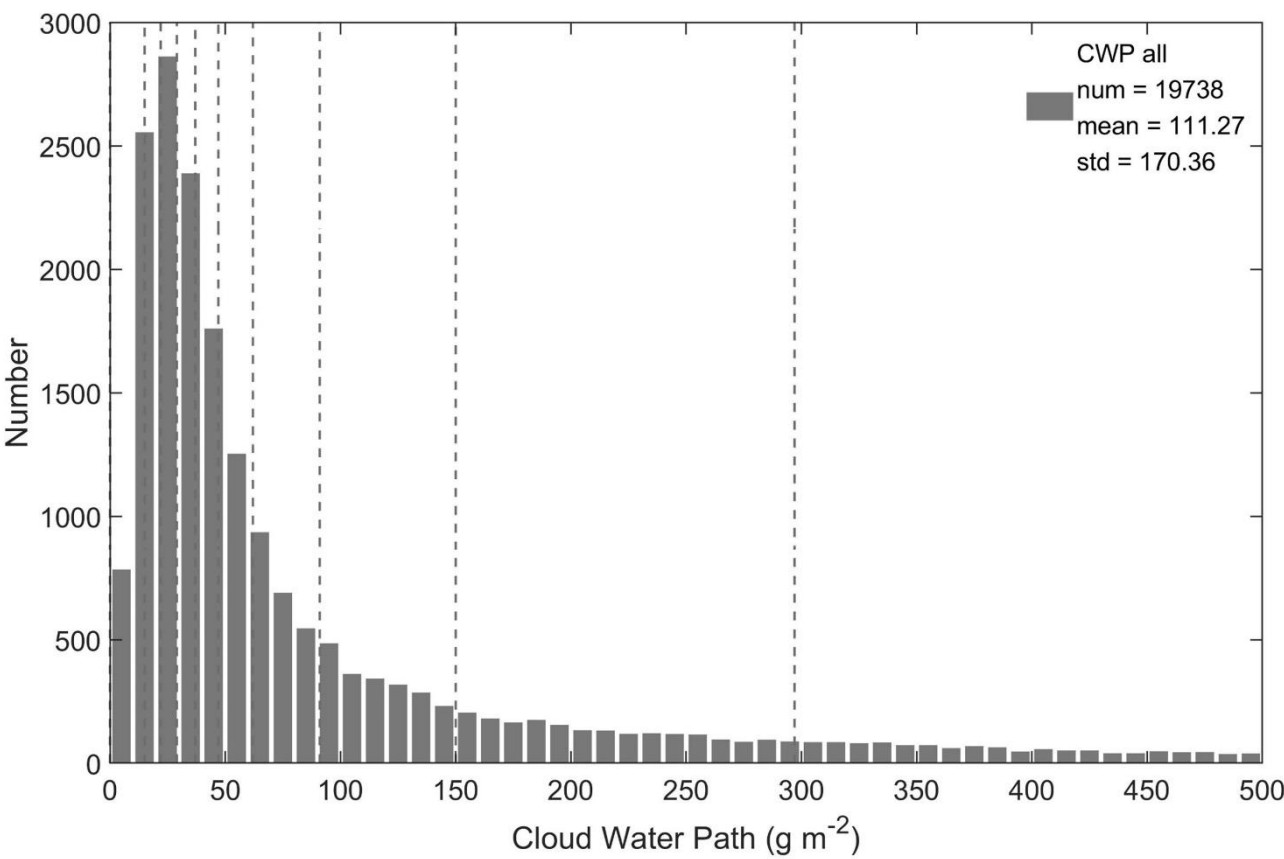

**Figure 2: Histogram of cloud water path (CWP) over northern Taiwan from Oct. 15 to Nov. 30, 2005–2017. The CWP is divided into 10 bins (10 % for each bin) indicated by dashed lines.**

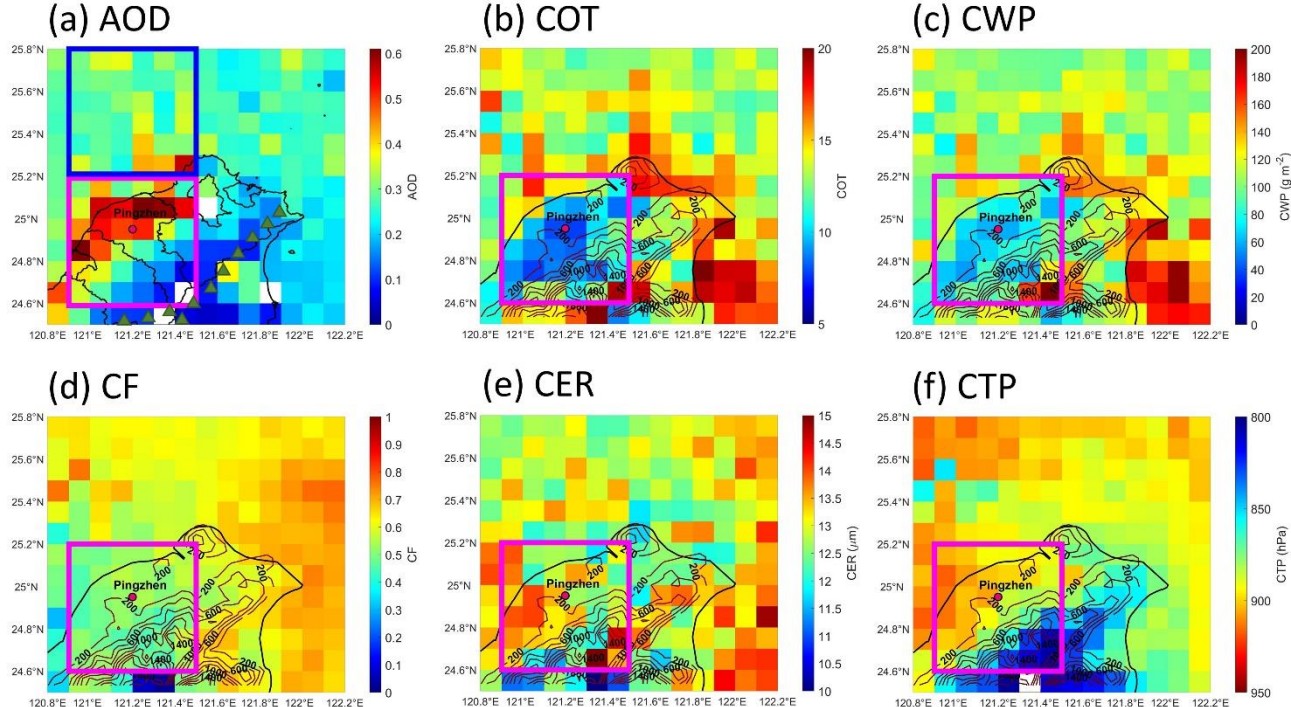

**Figure 3: Average (a) aerosol optical depth (AOD), (b) cloud optical thickness (COT), (c) cloud water path (CWP), (d) cloud fraction (CF), (e) cloud effective radius (CER), and (f) cloud top pressure (CTP) in warm clouds from October 15 to November 30, 2005–2017. The magenta box represents the main study area (24.6°–25.2° N, 120.9°–121.5° E) and the blue box in (a) is the remote area (25.2°–25.8° N, 120.9°–121.5° E). The green triangles in (a) represent the schematic of the Xueshan Range. The topography of north Taiwan is depicted with brown color contour lines (in meter) in (b)–(f).**



**Figure 4: The distribution of (a) temperature, (b) relative humidity, (c) rainfall, (d) wind direction, (e) wind speed, and (f) PM$_{2.5}$ hourly data from Pingzhen station from Oct. 15 to Nov. 30, 2005–2017. The gray bars are the distribution of all valid observations, the blue lines represent the clean days and the red lines represent the polluted days.**



**Figure 5: Multiyear (2005–2017) mean and standard deviation of temperature, relative humidity (RH), rainfall, PM2.5, and cloud effective radius (CER) in different cloud water path (CWP) bins.**



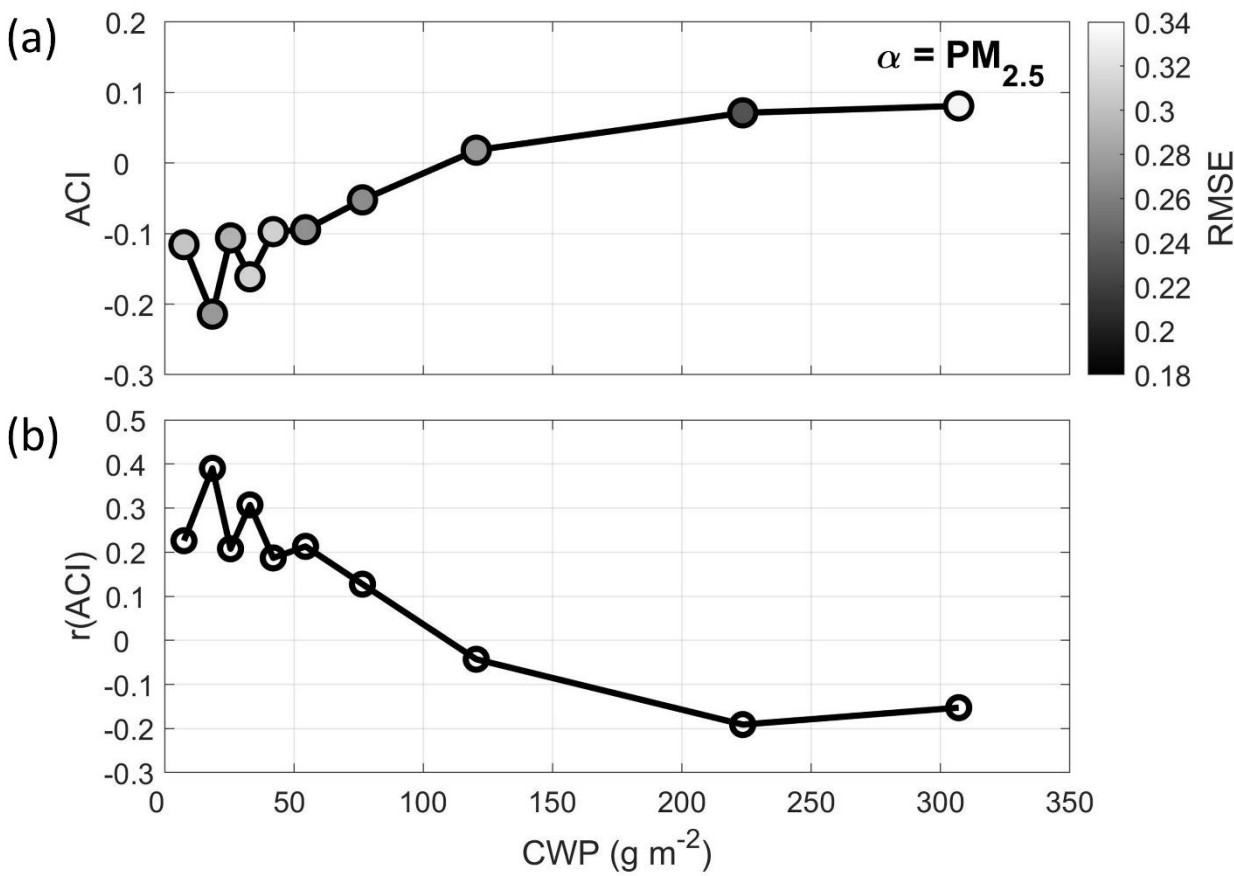

**Figure 6: (a) Aerosol cloud interaction (ACI) estimated values, computed for the cloud effective radius (CER) in the different CWP groups by applying the PM$_{2.5}$ as aerosol proxies. The shading in (a) represents the RMSE. (b) The correlation coefficients between PM$_{2.5}$ and CER are illustrated.**





**Figure 7: Difference in (a) COT, (b) CER, (c) CF, and (d) CTT between polluted days and clean days in group 9 (150 ≤ CWP < 297).**





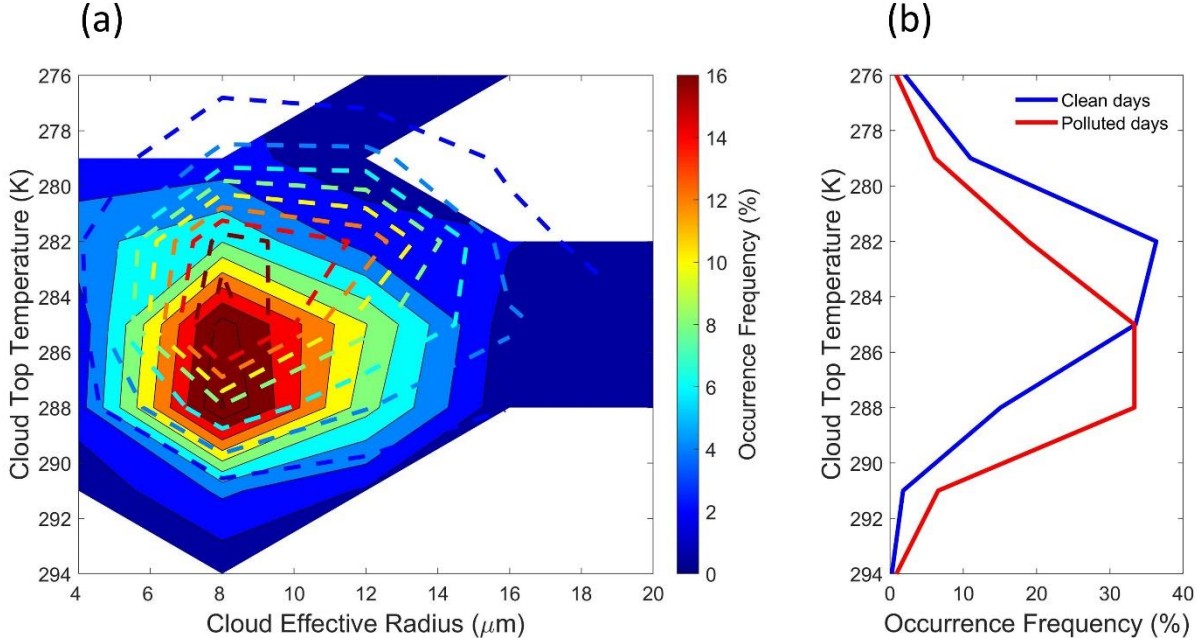

**Figure 8: Multiyear (2005–2017) (a) occurrence frequency (%) as a function of cloud top temperature (CTT) and cloud effective radius (CER). The color contour in the shaded area and the dashed lines denote the polluted and clean days, respectively. (b) The vertical profile of CTT occurrence frequencies for polluted and clean days are depicted with red and blue lines, respectively. Both in CWP group 9 (150 ≤ CWP < 297).**





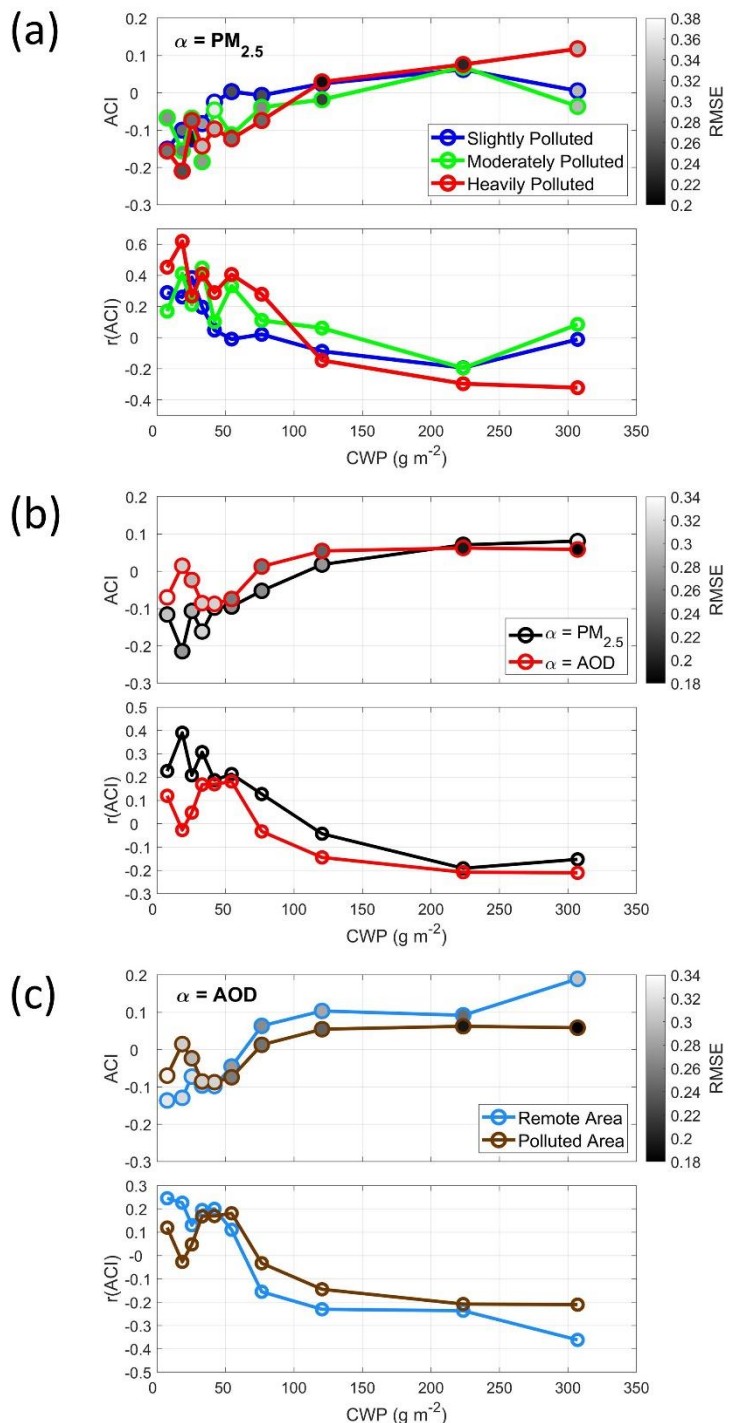

**Figure 9: Multiyear (2005–2017) ACI values with the RMSE (shaded) and the correlation coefficient among (a) different polluted levels, (b) different aerosol proxies, and (c) different polluted condition areas.**



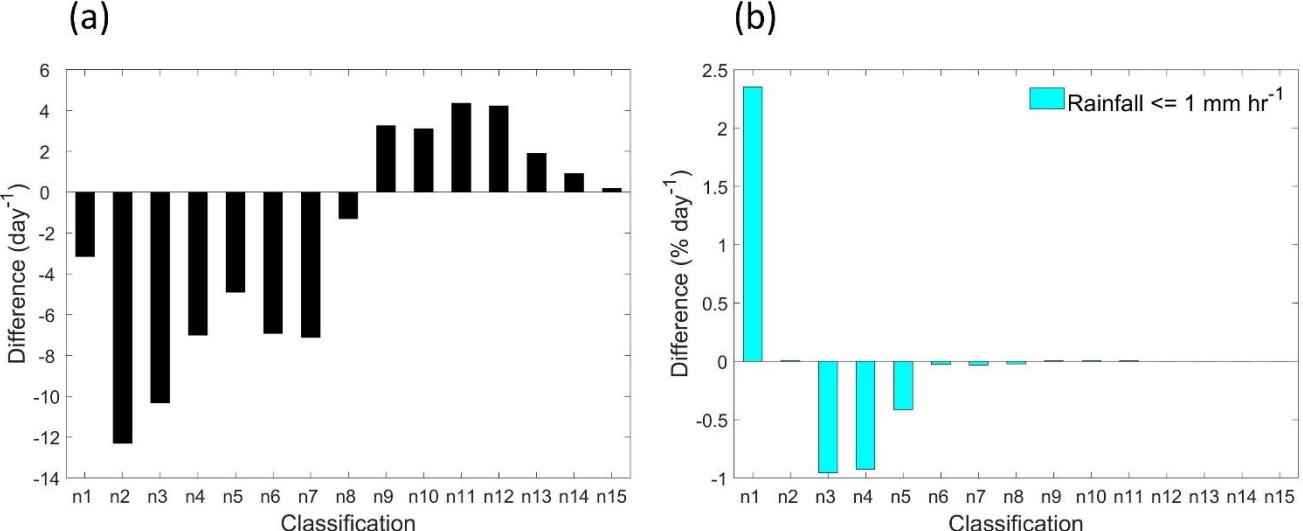

**Figure 10: Multiyear (2005–2017) (a) differences in the mean droplet number between polluted and clean days and (b) differences between polluted and clean days in the percentage of the cumulative droplet number distribution in different precipitation scenarios less than or equal to 1 mm h$^{-1}$. nX reflects different raindrop size bins. The n1 to n15 are, in order, 0.359, 0.455, 0.551, 0.656, 0.771, 0.913, 1.116, 1.331, 1.506, 1.665, 1.912, 2.259, 2.584, 2.869, and 3.198 mm.**

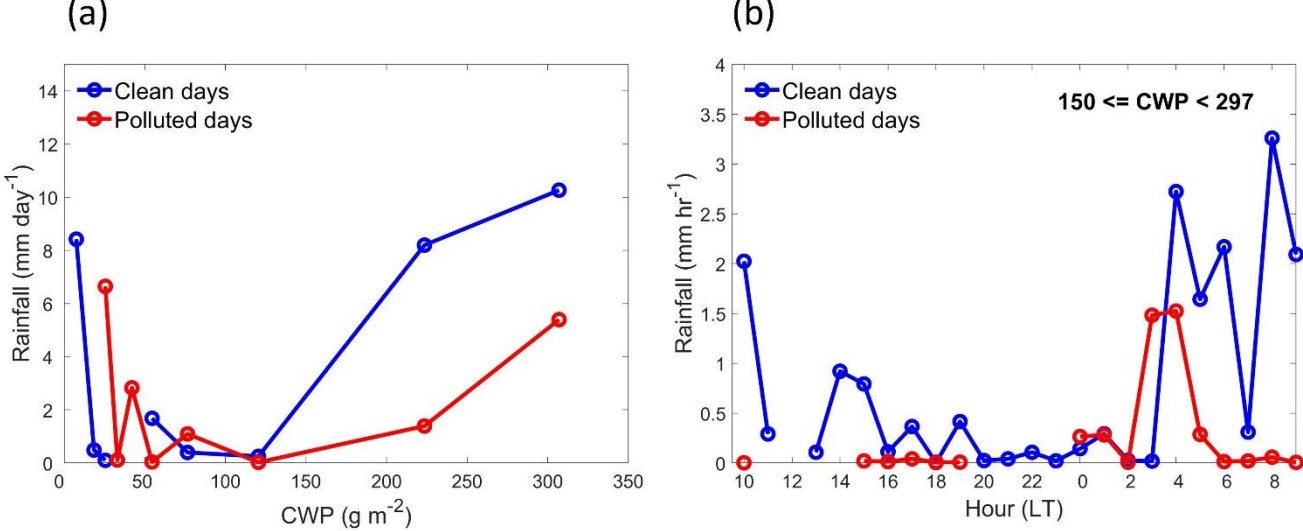

**Figure 11: Multiyear (2005-2017) (a) mean rainfall in different CWP groups calculated for clean and polluted days, and (b) 24-hours trend of average hourly-rainfall rate calculated for clean and polluted days and considered the CWP group 9 (150 ≤ CWP < 297) only. Rainfall analyses were performed from 10 am and the PM$_{2.5}$ data were averaged from 10 am to 2 pm as daily PM$_{2.5}$.**



**Table 1: MODIS aerosol and cloud products used in this study.**

| Product | Dataset | Acronym | Unit | Resolution |
|---|---|---|---|---|
| Aerosol (MYD04_L2, Collection 6) | Optical_Depth_Land_And_Ocean | AOD | | 10 km |
| Cloud (MYD06_L2, Collection 6) | Cloud_Effective_Radius | CER | μm | 1 km |
| | Cloud_Optical_Thickness | COT | | 1 km |
| | Cloud_Water_Path | CWP | $g\ m^{-2}$ | 1 km |
| | Cloud_Fraction | CF | | 5 km |
| | Cloud_Top_Pressure | CTP | hPa | 5 km |
| | Cloud_Top_Temperature | CTT | K | 5 km |
| | Cloud_Phase_Infrared | CPI | | 5 km |