# Peer review of "Aerosol impacts on warm-cloud microphysics and drizzle in a moderately polluted environment"

_Atmospheric Chemistry and Physics, 2020_

## Referee Comment (RC1) · Anonymous Referee #1 · 11 Sep 2020

General comments:

This study investigated aerosol impacts on cloud and precipitation over northern Taiwan using aerosol and cloud datasets from Aqua/MODIS and surface measurements. The authors showed statistical analysis including the susceptibility of cloud droplet effective radius (CER) to aerosols (ACI), correlations between CER and cloud-top temperature, and size distributions of rain drop to find some signatures of aerosol-induced changes to cloud and precipitation properties. Although the analysis results shown tend to be consistent with one another and thus appear to suggest the aerosol impacts on cloud and precipitation over the target region, most of the analysis approach and

the results shown, including the ACI analysis, relationships between rainfall and cloud water path, and CER-CTT joint statistics, are pretty much similar to what has already been done in a number of previous studies. I found no substantial novelty in materials included in the manuscript of its current form that deserves publication. Based on these evaluations, I cannot recommend the manuscript be considered for publication in Atmospheric Chemistry and Physics at least in its current form. One possible way for improving the overall study is to obtain a process-level insight into aerosol impacts on drizzle and precipitation exploiting the surface measurement of size distributions of rainfall, which might add some novelty to this study. Listed below are some specific points that (hopefully) might help the authors to re-construct their work in this direction for future potential submission of the revised manuscript.

Specific comments:

- A novel piece of material included in the manuscript is rain drop size distribution measured by the JWD disdrometer, which should provide useful observation-based information for process-level assessment of the aerosol indirect effect on precipitation, i.e. how precipitation processes are modulated by aerosols. I would suggest the authors to conduct more detailed analysis of the rain drop size distributions and their relationships to differing conditions of aerosols, rather than just showing the simple plot of Fig. 10. Such an analysis should offer size-dependent view of aerosol impact on drizzle and precipitation and thus more in-depth insight into microphysics of the aerosol indirect effect.

- The size-resolved precipitation analysis might also add new insight into the analysis shown in Fig. 11. The statistics shown in Fig. 11a is quite similar to those already shown by satellite statistics of Lebsock et al. (2008) and L'Ecuyer et al. (2009), except that the authors' plot shows the rainfall rate (in ordinate) based on surface measurement, contrary to probability of precipitation in the two previous studies. I would suggest the statistics shown in Fig. 11a be broken down into different bins of drop size to see how the cloud-to-precipitation process varies with aerosols and how it depends on particle size of drizzle and rain. Such an analysis might offer a new process-level insight into the aerosol-induced suppression of precipitation. The same approach could also be applied to the analysis of Fig. 11b to obtain a "size-resolved view" of the temporal trend of precipitation and its relationship to aerosols.

- The joint statistics between CER and CTT shown in Fig. 8 are hard to interpret in its current form. I guess that the authors like to claim different CTT-CER correlations between clean and polluted conditions in Fig. 8a, but the tendency looks quite ambiguous in the plot shown. I would suggest apply analysis methodology of Rosenfeld and colleagues (e.g. Rosenfeld and Lensky, 1998; Rosenfeld 2000) that plot the mean and variance of CER at each CTT bin separately for clean and polluted conditions. It might show more clearly what the authors want to illustrate.

- These analyses proposed above could then be combined to enable interpreting the traditional analysis such as the ACI and CER-CTT statistics in terms of size-resolved characteristics of precipitation processes. Such an analysis would connect some of the existing metrics of the aerosol indirect effect in the context of precipitation processes, which would bring a valuable progress in understanding aerosol impacts on cloud and precipitation.

Minor points:

- Page 5, Line 27: COT should have no unit. - Page 6, Line 30: radiuses -> radii - Page 7, Line 15: Does "cloud vertical profiles" mean CTT? It is not really the vertical profile but just a cloud-top temperature. - Figures 2, 5, 6, 9 and 11a: The horizontal axis for CWP should be logarithmic for at least some of the figures.

Reference:

Lebsock, M., G. L. Stephens, and C. Kummerow, 2008: Multisensor satellite observations of aerosol effects on warm clouds. J. Geophys. Res., 113, D15205, doi:10.1029/2008JD009876.

L'Ecuyer, T. S., W. Berg, J. Haynes, M. Lebsock, and T. Takemura, 2009: Global observations of aerosol impact on precipitation occurrence in warm maritime clouds. J. Geophys. Res., 114, D09211, doi:10.1029/2008JD011273.

Rosenfeld, D., and I. M. Lensky, 1998: Satellite-based insights into precipitation formation processes in continental and maritime convective clouds. Bull. Ame. Meteorol. Soc., 79, 2457-2476.

Rosenfeld, D., 2000: Suppression of rain and snow by urban and industrial air pollution. Science, 287, 1793-1796.

---

## Referee Comment (RC2) · Anonymous Referee #2 · 17 Sep 2020

The authors present a nice, if perhaps a little over-extensive, study looking at in situ and some satellite measurements in an urban and complex setting. While the analysis presented here in some cases is not new, the data analysis of in situ data is hard and different and the analysis warrants publishing to add to our growing knowledge of aci. I find some of the discussion of adjustments overly assertive of causality, which the authors cannot show empirically. These regions need to be trimmed to report on findings without asserting a causal connection, or the authors should perform modelling of the region where they can make some advances to understanding the direction of causality in what their observations are doing. While I acknowledge that many studies utilize CER to calculate aci, I would suggest using Nd, which the authors have already

calculated to provide a complimentary calculation that may be more relevant to more recent studies. The authors may also wish to say a few words about why PM2.5 may be a good CCN and need to address near-cloud aerosol swelling in the text, which makes the direction of causality even more difficult to infer. The use of the rain size distribution is a good way to approach this problem.

Another way the authors might want to consider looking at this is performing the same analysis in their paper, but instead of sorting clean/polluted sorting by atmospheric advection from the east or west. This might reveal the underlying meteorological signal that will covary with aerosol. This result can be used to say 'on days when the dominant weather pattern is such, but there is unusually little aerosol then the clouds do this'.

P1 L15: I am not sure what this sentence is getting at- is the human activity causing low cloud?

P2 L17: You should discuss spurious correlation between AOD and cloud properties as shown in (Christensen et al. 2017; Twohy et al. 2009).

P2 L31: I might say weakly constrained (Bellouin et al. 2020).

P3 L3: What does largely dominant mean? Relative to what?

P3L6: It seems like it might be good to discuss this in the context of the current synthesis report on aci (Bellouin et al. 2020).

P4L13: So AOD was only retrieved when AOD was visible? It seems like all periods with cloud should be zeroed out since there might be AOD below cloud that is not being counted.

P4L19: What is this based on? Afternoon aerosol should be able to affect afternoon clouds.

P5 L1: This would be more reliably at a constant CWP if cloud droplet number concentration (Nd) was used instead of CER and binning by CWP (Grosvenor et al. 2018).

Any inferred aci will be a function of binning decisions.

P7L7: This is a nice comparison to previous studies. Please consider summarizing in a figure.

P7L13: This is not a robust piece of analysis. Differing PM2.5 is likely a function of atmospheric state (air masses moving from the west for instance) and this is likely to do more to CF and COT than aci.

P8L6 please comment on the unintuitive diagnosed stronger aci in more polluted clouds. A lot of studies point to stronger aci in more pristine clouds(Carslaw et al. 2013). Again, this may be a function of binning, which is also going to select for clouds in an atmospheric regime.

P9L4: Or the cleaner days could be occurring because of rain scavenging aerosol. Unfortunately in an emprical study such as this you can't make causal statements. However, the high temporal resolution of ground data used here might allow for some sort of time evolution analysis that could show causality.

P9L22: Please note that precipitation reduction is often a function of model parameterization.

Bellouin, N., and Coauthors, 2020: Bounding Global Aerosol Radiative Forcing of Climate Change. Rev. Geophys., 58. Carslaw, K. S., and Coauthors, 2013: Large contribution of natural aerosols to uncertainty in indirect forcing. Nature, 503, 67-71. Christensen, M. W., and Coauthors, 2017: Unveiling aerosol–cloud interactions – Part 1: Cloud contamination in satellite products enhances the aerosol indirect forcing estimate. Atmos. Chem. Phys., 17, 13151-13164. Grosvenor, D. P., and Coauthors, 2018: Remote Sensing of Droplet Number Concentration in Warm Clouds: A Review of the Current State of Knowledge and Perspectives. Rev. Geophys., 56, 409-453. Twohy, C. H., J. A. Coakley, and W. R. Tahnk, 2009: Effect of changes in relative humidity on aerosol scattering near clouds. J Geophys Res-Atmos, 114, n/a-n/a.

---

## Author Comment (AC1) · 1 Dec 2020

We greatly appreciate the constructive review from the referee that has improved the quality of our manuscript. We have considered each comment carefully and revised our manuscript accordingly to address the issues raised. Below we address each comment point by point. Reviewer comments are marked as black, our response as blue and changes to the manuscript as red.

This study investigated aerosol impacts on cloud and precipitation over northern Taiwan using aerosol and cloud datasets from Aqua/MODIS and surface measurements. The authors showed statistical analysis including the susceptibility of cloud droplet effective radius (CER) to aerosols (ACI), correlations between CER and cloud-top temperature, and size distributions of rain drop to find some signatures of aerosol-induced changes to cloud and precipitation properties. Although the analysis results shown tend to be consistent with one another and thus appear to suggest the aerosol impacts on cloud and precipitation over the target region, most of the analysis approach and the results shown, including the ACI analysis, relationships between rainfall and cloud water path, and CER-CTT joint statistics, are pretty much similar to what has already been done in a number of previous studies. I found no substantial novelty in materials included in the manuscript of its current form that deserves publication. Based on these evaluations, I cannot recommend the manuscript be considered for publication in Atmospheric Chemistry and Physics at least in its current form. One possible way for improving the overall study is to obtain a process-level insight into aerosol impacts on drizzle and precipitation exploiting the surface measurement of size distributions of rainfall, which might add some novelty to this study. Listed below are some specific points that (hopefully) might help the authors to re-construct their work in this direction for future potential submission of the revised manuscript.

We really appreciate and agree with these suggestions and comments from the referee. We have strengthened the analysis, in particular, the process-level insight into aerosol impacts on drizzle and precipitation by exploiting the surface measurement of rainfall size-distributions (lines: 257-283). As suggested, the analysis of ACI and CER-CTT statistics in terms of size-resolved characteristics of precipitation processes were included to support the discussion (lines: 206-215). We have addressed the specific comments in the sections below and made the revisions to the manuscript accordingly.

In addition, we believe our target region may be unique and stand out from other previous studies. First, the study area is located in the northwest Pacific Ocean where there has been much attention on aerosol transportation, as well as aerosol-cloud interactions from the literature (Tsay et al., 2016; Dong et al., 2019). However, observational-based studies are still lacking in this region. Second, this study integrates long-term satellite and surface measurements to assess ACI over a moderately polluted environment with complex terrain. Although the overall result appears similar to previous studies, it has important implications for the crucial role of cloud microphysics on the water cycle/resources in subtropical East Asia environment.

**Specific comments:**

- A novel piece of material included in the manuscript is rain drop size distribution measured by the JWD disdrometer, which should provide useful observation-based information for process-level assessment of the aerosol indirect effect on precipitation, i.e. how precipitation processes are modulated by aerosols. I would suggest the authors to conduct more detailed analysis of the rain drop size distributions and their relationships to differing conditions of aerosols, rather than just showing the simple plot of Fig. 10. Such an analysis should offer size-dependent view of aerosol impact on drizzle and precipitation and thus more in-depth insight into microphysics of the aerosol indirect effect.

Many thanks for this suggestion. We have added a more detailed analysis of the raindrop size distributions and the aerosol impact on drizzle and precipitation via the aerosol indirect effect. The paragraph has now been rewritten (lines: 257-283) and revised the original Fig. 10 to Fig. 10 and Fig. 11 as below:

Figure 10a shows the number of sample occurrences under different raindrop size classifications for clean and polluted days. The sample number (days) was significantly higher for clean conditions, suggesting rainfall was more common on clean days than on polluted days. We further calculated the minute-averaged droplet number in each raindrop size classification for polluted and clean days. Higher populations of raindrops were observed from bins n1 to n4, with the peak in bin n2 for both clean and polluted days (Fig. 10b). The difference is plotted in Fig. 10c. The results

illustrate (Fig. 10c) that during polluted days, the droplet numbers appear lower for the smaller raindrop bins ($\leq$ n8) compared to clean days and higher for the larger raindrop bins ($>$ n8). A significant reduction in droplet number (decreased from 68 min$^{-1}$ on clean days to 56 min$^{-1}$ on polluted days) was observed in the n2 bin, corresponding to a reduction in drizzle. Our preliminary findings suggest that CCN may have competing effects (Ghan et al., 1998) on water uptake under aerosol-laden air and cloud water content-limited conditions, which would alter the precipitation processes.

[Figure]

Figure 10: Multiyear (2005-2017) (a) JWD sample number of days in each raindrop size bin, (b) mean droplet number per minute for clean and polluted days and (c) The differences in the mean droplet number between polluted and clean days. nX reflects different raindrop size bins. The mean droplet size for n1 to n15 are, in order, 0.359, 0.455, 0.551, 0.656, 0.771, 0.913, 1.116, 1.331, 1.506, 1.665, 1.912, 2.259, 2.584, 2.869, and 3.198 mm.

To investigate the aerosol impacts on the change in droplet size, the cumulative number distribution of each raindrop size for clean and polluted days was calculated. We then normalized the data by computing the percentage of droplet numbers in each raindrop size class to the total number and the difference between polluted and clean days was defined by Eq. (2).

$$\text{nX Difference } (\% \ min^{-1}) = \frac{\sum_{i=1}^{dp} nX_i}{\sum_{X=1}^{b} \sum_{i=1}^{dp} nX_i} \times 100 \ \% \ - \frac{\sum_{i=1}^{dc} nX_i}{\sum_{X=1}^{b} \sum_{i=1}^{dc} nX_i} \times 100 \ \%, \tag{2}$$

where nX represents different raindrop size bins and b reflects the number of bins, b = 1-20; dp

and dc represent the number of polluted and clean days respectively. The results are similar with Fig. 10c; the droplet numbers, on polluted days compared to clean days, appear lower for the smaller raindrop bins ($\leq$ n5) and higher for the larger raindrop bins ($>$ n5) (Fig. 11a). To investigate the aerosol impacts on light rain, we created a similar plot as Fig. 11a but only considered precipitation less than or equal to 1 mm h$^{-1}$, as shown in Fig. 11b. Our statistics for the droplet number concentration indicated that raindrop occurrence at n1 and n2 (i.e. drizzle) accounted for over 50 % on both polluted and clean days (not shown here) (shown as Fig. R1 in this response, but not shown in the revised manuscript), indicating that drizzle drops were a common raindrop type when rainfall was $\leq$ 1 mm h$^{-1}$. We determined that when rainfall was $\leq$ 1 mm h$^{-1}$, polluted days accounted for a more significant proportion of n1 and n2 than clean days (especially in the raindrop size distribution n1, which accounted for 2.3 %) (Fig. 11b). On the other hand, a decreased proportion of n3 to n8 was observed during polluted days, as compared with clean days. These results indicate that if precipitation is lower than or equal to 1 mm h$^{-1}$ (i.e. light rain), abundant CCN drives raindrops to move towards smaller drop sizes, which increases the appearance of drizzle drops.

[Figure]

Figure 11: Multiyear (2005-2017) differences between polluted and clean days as percentages of the cumulative droplet number distribution for (a) all data and (b) the data with precipitation less than or equal to 1 mm h$^{-1}$. nX reflects different raindrop size bins as listed in Fig. 10.

[Figure]

Figure R1: Multiyear (2005-2017) cumulative droplet number distribution for the JWD data for precipitation less than or equal to 1 mm h[-1] on clean and polluted days. nX reflects different raindrop size bins as specified in Fig. 10.

- The size-resolved precipitation analysis might also add new insight into the analysis shown in Fig. 11. The statistics shown in Fig. 11a is quite similar to those already shown by satellite statistics of Lebsock et al. (2008) and L'Ecuyer et al. (2009), except that the authors' plot shows the rainfall rate (in ordinate) based on surface measurement, contrary to probability of precipitation in the two previous studies. I would suggest the statistics shown in Fig. 11a be broken down into different bins of drop size to see how the cloud-to-precipitation process varies with aerosols and how it depends on particle size of drizzle and rain. Such an analysis might offer a new process-level insight into the aerosol-induced suppression of precipitation. The same approach could also be applied to the analysis of Fig. 11b to obtain a "size-resolved view" of the temporal trend of precipitation and its relationship to aerosols.

Many thanks for this suggestion. We followed the suggestion and binned the rainfall data into drop size to study how the cloud-to-precipitation process varies with aerosol concentration and how it depends on the particle size of drizzle and rain. We divided the droplet bins into three groups: n1-n20, n1-n2, and n3-n20, representing all droplets, drizzle drops, and raindrops, respectively. We calculated the minute-averaged droplet number in each group of bins. The results shown in Fig. R2a, b, c demonstrate that the mean droplet number difference between polluted and clean days varies greatly between CWP groups 1–7, which may be due to the smaller sample number in each CWP group. However, whether drizzle drops or raindrops, the mean droplet number on clean days consistently exhibited higher values in CWP groups 8–10 compared with polluted days and increased with increasing CWP. In CWP group 9 ($150 \leq$ CWP $< 297$), the mean droplet number on polluted days (12 min$^{-1}$) was lower by 38 min$^{-1}$ compared with clean days (50 min$^{-1}$) when considering all droplets (Fig. R2a).

Figure R2d, e, f shows the 24-hour mean droplet number trends for CWP group 9 ($150 \leq$ CWP $< 297$) on clean and polluted days, providing insights on the effect of aerosols on cloud lifetime. On clean days, when considering all droplets (n1-n20), the droplet number was larger than 50 min$^{-1}$ except at 12:00, 20:00-23:00 and 02:00-03:00, whereas few droplets were observed during daytime on polluted days, and a droplet number greater than 50 min$^{-1}$ registering only sporadically after 23:00. Considering raindrops (n3-n20), there was a notably larger droplet number observed after 03:00 (Fig. R2f). This may have been caused by high aerosol loading suppressing the precipitation in the daytime, delaying rainfall occurrence and in turn increasing the droplet number of larger raindrops in the early morning.

The above-mentioned results are in agreement with our revised manuscript discussing aerosol effects on precipitation (in Sect. 3.4), and suggesting precipitation might be suppressed and delayed under high aerosol loading. To avoid confusion for readers, this revised manuscript does not include the supplementary analysis described above.

[Figure]

Figure R2: Multiyear (2005-2017) mean droplet number for (a) all droplets, (b) drizzle drops, and (c) raindrops in different CWP groups calculated for clean and polluted days. Hourly trend of mean droplet number for (d) all droplets, (e) drizzle drops, and (d) raindrops calculated for clean and polluted days when considering CWP group 9 (150 ≤ CWP < 297) only.

- The joint statistics between CER and CTT shown in Fig. 8 are hard to interpret in its current form. I guess that the authors like to claim different CTT-CER correlations between clean and polluted conditions in Fig. 8a, but the tendency looks quite ambiguous in the plot shown. I would suggest apply analysis methodology of Rosenfeld and colleagues (e.g. Rosenfeld and Lensky, 1998; Rosenfeld 2000) that plot the mean and variance of CER at each CTT bin separately for clean and polluted conditions. It might show more clearly what the authors want to illustrate.

Thank you for the valuable suggestion. We now reference the analysis methodology of Rosenfeld (2000), and plot the mean and one standard deviation of CER at each CTT bin. The paragraph has been rephrased as below (lines: 206-215 in the revised manuscript):

The relationship between CTT and CER and aerosols was studied in further detail. Figure 8 displays CWP group 9 (150 ≤ CWP < 297) results of the corresponding CTT-CER relationship and the occurrence frequency (%) of the CTT on clean and polluted days. On clean days, the mean CER increased from 10.7 to 12.7 μm as CTT decreased from 291 to 279 K, indicating an inverse relationship over much of the CTT range. This phenomenon could be caused by the onset of water cloud generation during strong updrafts, i.e. droplet size increases during air parcel expansion in an adiabatic process (Saito et al., 2019). However, on polluted days, as CTT lowered, the mean CER decreased; at CTT from 291 to 279 K, the CER decreased from 10.8 to 9.1 μm. Figure 8b shows that CTT exhibited a higher occurrence frequency between 288 and 285 K on polluted days, whereas clean days had a higher frequency of CTT between 285 and 282 K. These results suggest that abundant aerosols activated higher concentrations of CCN near the surface, which tends to form more low-level clouds with smaller cloud droplet size.

[Figure]

Figure 8: Multiyear (2005–2017) (a) cloud top temperature (CTT)-cloud effective radius (CER) relationship. Plotted are the mean (solid line) and one standard deviation (dashed line) of the CER for each 3 K interval, and (b) Frequency of occurrence of the CTT. Clean and polluted days are depicted with blue and red lines, respectively. Both (a) and (b) are constrained to CWP group 9 $(150 \leq CWP < 297)$.

- These analyses proposed above could then be combined to enable interpreting the traditional analysis such as the ACI and CER-CTT statistics in terms of size-resolved characteristics of precipitation processes. Such an analysis would connect some of the existing metrics of the aerosol indirect effect in the context of precipitation processes, which would bring a valuable progress in understanding aerosol impacts on cloud and precipitation.

Thanks for the comments. Complementing the revisions mentioned above, the conclusions have been rephrased as (lines: 313-331 in the revised manuscript):

We used surface $PM_{2.5}$ mass concentration data as aerosol proxy to study the aerosol impacts on clouds and precipitation. According to $PM_{2.5}$ concentration level, the data was split into clean and polluted days. The analysis of aerosol effects on clouds indicated that in CWP group 9 $(150 \leq$

CWP < 297), the average COT in the main research area increased by 9.53, CER decreased by 2.77 μm, CF increased by 0.07, and CTT decreased by 1.28 K on polluted days compared with clean days. According to the aerosol indirect effect, polluted atmospheric conditions are connected with clouds characterized by lower CER, CTP, and larger CF and COT, which our results further support. Regarding the vertical distribution, our evidence shows that excess aerosols produced more liquid particles at lower altitude and inhibited the cloud droplet size under polluted conditions. Moreover, the effects of aerosol on cloud microphysics in polluted (i.e. land) and remote (i.e. ocean, less polluted) areas were investigated in CWP group 9, the ACI value of the remote area was 0.09, and the polluted area was 0.06. The ACI value in the remote area was larger than in the polluted area, indicating that clouds in the remote area were more sensitive to aerosol indirect effects.

Our analysis shows that precipitation might be suppressed and delayed under high aerosol loading. The observational data shows higher aerosol concentration redistributed cloud water to more numerous and smaller droplets under a constant liquid water content, reducing collision–coalescence rates, which further suppressed the precipitation and delayed rainfall duration. Our results are consistent with the cloud lifetime effect. Finally, we combined the observation of raindrop size distribution to complete the story of aerosol-cloud-precipitation interactions. As a result, on polluted days compared to clean days, droplet numbers decreased for smaller droplets bins but increased for larger droplets. However, when we looked into the light rain ($\leq 1$ mm h$^{-1}$) category, high concentration of aerosols drove raindrops towards smaller droplet sizes and increased the appearance of drizzle drops.

**Minor points:**

- Page 5, Line 27: COT should have no unit.

Thank you for correcting our mistakes. The sentence has been rephrased as (lines: 154-156):

The mean CWP, CF and CER in our study area ranged from 60–120 g m$^{-2}$, 0.6–0.7, and 13–14.5 μm, respectively. **COT was usually around 10** and most of the CTP was higher than 850 hPa, suggesting low-level clouds (e.g., warm, thin, and broken clouds).

- Page 6, Line 30: radiuses -> radii

Thank you for the correction. The sentence has been rephrased as (lines: 185-186):

The negative correlation for these groups indicates an aerosol indirect effect (i.e. an increase in aerosols cause cloud droplet **radii** to become smaller under a fixed water content).

- Page 7, Line 15: Does "cloud vertical profiles" mean CTT? It is not really the vertical profile but just a cloud-top temperature.

We agree with the reviewer's insight that CTT is not really the vertical profile but just a cloud top temperature. The paragraph has been rephrased as (lines: 206-215):

The relationship between CTT and CER and aerosols was studied in further detail. Figure 8 displays CWP group 9 ($150 \leq CWP < 297$) results of the corresponding CTT-CER relationship and the occurrence frequency (%) of the CTT on clean and polluted days. On clean days, the mean CER increased from 10.7 to 12.7 µm as CTT decreased from 291 to 279 K, indicating an inverse relationship over much of the CTT range. This phenomenon could be caused by the onset of water cloud generation during strong updrafts, i.e. droplet size increases during air parcel expansion in an adiabatic process (Saito et al., 2019). However, on polluted days, as CTT lowered, the mean CER decreased; at CTT from 291 to 279 K, the CER decreased from 10.8 to 9.1 µm. Figure 8b shows that CTT exhibited a higher occurrence frequency between 288 and 285 K on polluted days, whereas clean days had a higher frequency of CTT between 285 and 282 K. These results suggest that abundant aerosols activated higher concentrations of CCN near the surface, which tends to form more low-level clouds with smaller cloud droplet size.

- Figures 2, 5, 6, 9 and 11a: The horizontal axis for CWP should be logarithmic for at least some of the figures.

Thank you for the suggestion. We re-plotted the horizontal axis in logarithmic CWP and CWP group as shown below in Fig. R3. Although they have similar patterns, after our internal discussion, we decided to re-plot the original Fig. 6 and Fig. 9 with an x-axis of CWP group in the revised

manuscript.

[Figure]

Figure R3: Aerosol cloud interaction estimated values vs. three different CWP variables (a) CWP, (b) logarithmic CWP, and (c) CWP group number.

[Figure]

Figure 6: (a) Aerosol cloud interaction (ACI) estimated values, computed for the cloud effective radius (CER) in the different CWP groups by applying $PM_{2.5}$ concentrations as aerosol proxies. The shading in (a) represents the RMSE. (b) The correlation coefficients between $PM_{2.5}$ and CER are illustrated.

[Figure]

Figure 9: Multiyear (2005–2017) ACI values with the RMSE (shaded) and the correlation coefficient among (a) different polluted levels, (b) different aerosol proxies, and (c) different polluted condition areas.

**Reference is cited in the response letter:**

Dong, B., Wilcox, L. J., Highwood, E. J., and Sutton, R. T.: Impacts of recent decadal changes in Asian aerosols on the East Asian summer monsoon: roles of aerosol–radiation and aerosol–cloud interactions, Climate Dynamics, 53, 3235-3256, 2019.

Rosenfeld, D.: Suppression of rain and snow by urban and industrial air pollution, Science, 287, 1793-1796, 2000.

Tsay, S.-C., Maring, H. B., Lin, N.-H., Buntoung, S., Chantara, S., Chuang, H.-C., Gabriel, P. M., Goodloe, C. S., Holben, B. N., and Hsiao, T.-C.: Satellite-surface perspectives of air quality and aerosol-cloud effects on the environment: An overview of 7-SEAS/BASELInE, Aerosol and Air Quality Research, 16, 2581-2602, 2016.

**New reference is added in the revised manuscript:**

Ghan, S. J., Guzman, G., and Abdul-Razzak, H.: Competition between sea salt and sulfate particles as cloud condensation nuclei, Journal of the atmospheric sciences, 55, 3340-3347, 1998.

---

## Author Comment (AC2) · 1 Dec 2020

We greatly appreciate the constructive review from the referee that has improved the quality of our manuscript. We have considered each comment carefully and revised our manuscript accordingly to address the issues raised. Below we address each comment point by point. Reviewer comments are marked as black, our response as blue and changes to the manuscript as red.

The authors present a nice, if perhaps a little over-extensive, study looking at in situ and some satellite measurements in an urban and complex setting. While the analysis presented here in some cases is not new, the data analysis of in situ data is hard and different and the analysis warrants publishing to add to our growing knowledge of aci.

We appreciate the reviewer for recognizing the value of this work. Specific points raised by the reviewer have been carefully considered and addressed in the following replies. In particular, much of the related revisions are focused on aerosol and cloud properties used in ACI and the discussion of the relevant results. We have addressed each comments in the sections below and made revisions to the manuscript accordingly.

I find some of the discussion of adjustments overly assertive of causality, which the authors cannot show empirically. These regions need to be trimmed to report on findings without asserting a causal connection, or the authors should perform modelling of the region where they can make some advances to understanding the direction of causality in what their observations are doing.

We thank the reviewer for pointing out this concern. We also agree that a modeling study may enhance our knowledge of the causality; however, this would extend us beyond our current capacity. Instead of a modeling component, we have revisited our observational data, with particular focus on CER-CTT statistics and raindrop size distribution analysis, allowing us to obtain a process-level insight into aerosol impacts on drizzle and precipitation. In the revised manuscript, we have added two figures about aerosol effects on precipitation. Figure 10 shows the multiyear (2005-2017) JWD sample number (days), mean droplet number per minute and the differences between polluted and clean days of the mean droplet number in each bin. The droplet number in the n2 bin was significantly lower on polluted days, indicating less drizzle in that

condition. Fig. 11 shows differences between polluted and clean days in the percentage of the cumulative droplet number distribution for (a) all data and (b) data with precipitation less than or equal to 1 mm h-1. The results using all data are similar with Fig. 10c; the droplet numbers appear lower for the smaller raindrop bins ( $\leq$  n5) on polluted days compared to clean days, and higher for the larger raindrop bins (> n5) (Fig. 11a). When precipitation is lower than or equal to 1 mm h-1 (i.e. light rain), abundant CCN drives raindrops towards smaller drop sizes, effectively increasing the number of drizzle drops (Fig. 11b).

Figure 10: Multiyear (2005-2017) (a) JWD sample number of days in each raindrop size bin, (b) mean droplet number per minute for clean and polluted days and (c) The differences in the mean droplet number between polluted and clean days. nX reflects different raindrop size bins. The mean droplet size for n1 to n15 are, in order, 0.359, 0.455, 0.551, 0.656, 0.771, 0.913, 1.116, 1.331, 1.506, 1.665, 1.912, 2.259, 2.584, 2.869, and 3.198 mm.

Figure 11: Multiyear (2005-2017) differences between polluted and clean days as percentages of the cumulative droplet number distribution for (a) all data and (b) the data with precipitation less than or equal to 1 mm  $h^{-1}$ . nX reflects different raindrop size bins as listed in Fig. 10.

In addition, we rewrote some paragraphs of findings by adding references rather than asserting a causal connection. The summary paragraph in Sect. 3.4 is rephrased as (lines: 297-303):

Although the existence of an aerosol effect on cloud lifetime is still widely disputed (Small et al., 2009; Stocker, 2014), our preliminary results show that precipitation might be suppressed and delayed under high aerosol loading. Combined with the results from Sect. 3.2, the process in the aerosol-cloud-precipitation interactions is consistent with the cloud lifetime effect. The presence of aerosols enhances the concentration of condensation nuclei under a fixed water content, which increases the cloud droplet number, redistributes cloud water to more numerous and smaller droplets, reducing collision–coalescence rates, which in turn suppresses precipitation and delays rainfall occurrence (i.e. the cloud lifetime effect (Albrecht, 1989; Pincus and Baker, 1994; Lohmann and Feichter, 2005)).

And a portion of the conclusions has been rephrased as (lines: 313-331):

We used surface PM2.5 mass concentration data as aerosol proxy to study the aerosol impacts on clouds and precipitation. According to PM2.5 concentration level, the data was split into clean and polluted days. The analysis of aerosol effects on clouds indicated that in CWP group 9 ( $150 \le CWP < 297$ ), the average COT in the main research area increased by 9.53, CER decreased by 2.77 µm, CF increased by 0.07, and CTT decreased by 1.28 K on polluted days compared with clean days. According to the aerosol indirect effect, polluted atmospheric conditions are connected with clouds characterized by lower CER, CTP, and larger CF and COT, which our results further support. Regarding the vertical distribution, our evidence shows that excess aerosols produced more liquid particles at lower altitude and inhibited the cloud droplet size under polluted conditions. Moreover, the effects of aerosol on cloud microphysics in polluted (i.e. land) and remote (i.e. ocean, less polluted) areas were investigated in CWP group 9, the ACI value of the remote area was 0.09, and the polluted area was 0.06. The ACI value in the remote area was larger than in the polluted area, indicating that clouds in the remote area were more sensitive to aerosol indirect effects.

Our analysis shows that precipitation might be suppressed and delayed under high aerosol loading. The observational data shows higher aerosol concentration redistributed cloud water to more numerous and smaller droplets under a constant liquid water content, reducing collision– coalescence rates, which further suppressed the precipitation and delayed rainfall duration. Our results are consistent with the cloud lifetime effect. Finally, we combined the observation of raindrop size distribution to complete the story of aerosol-cloud-precipitation interactions. As a result, on polluted days compared to clean days, droplet numbers decreased for smaller droplets bins but increased for larger droplets. However, when we looked into the light rain ( $\leq 1 \text{ mm h}^{-1}$ ) category, high concentration of aerosols drove raindrops towards smaller droplet sizes and increased the appearance of drizzle drops.

While I acknowledge that many studies utilize CER to calculate aci, I would suggest using  $N_d$ , which the authors have already calculated to provide a complimentary calculation that may be more relevant to more recent studies.

Thank you for the suggestion. We agree with the reviewer's insight that cloud droplet number concentration ( $N_d$ ) calculation may be more relevant. Grosvenor et al. (2018) indicated that  $N_d$  is of central interest to improve the understanding of cloud microphysics and for quantifying the effective radiative forcing by aerosol-cloud interactions. However, current standard satellite retrievals do not operationally provide  $N_d$ . It can be inferred from retrievals of cloud optical depth (COD), cloud droplet effective radius (CER) and cloud top temperature, but errors propagated from passive retrievals of COD and CER will generate uncertainties in the subsequently derived  $N_d$  (Grosvenor et al., 2018); thus, we currently retain the calculation of ACI by using CER.

The authors may also wish to say a few words about why  $PM_{2.5}$  may be a good CCN and need to address near-cloud aerosol swelling in the text, which makes the direction of causality even more difficult to infer. The use of the rain size distribution is a good way to approach this problem.

Thank you for the suggestion. In the revised manuscript, we have added a relevant sentence to specify the  $PM_{2.5}$  characteristics that we considered for using it as a suitable proxy of CCN (lines: 110-113):

The composition of  $PM_{2.5}$  in East Asia is usually dominated by carbonaceous species and water soluble ions, including  $SO_4^{2-}$ ,  $NH_4^+$ , and  $NO_3^-$  (Xu et al., 2012), which are important in determining the hygroscopicity of aerosols (Shen et al., 2009). Thus, based on these suitable characteristics and the lack of measured CCN in this study, we used  $PM_{2.5}$  as a proxy for CCN concentrations.

Li et al. (2017) used PM2.5 measurements to represent aerosol loading under cloudy conditions and showed significant negative relationships between cloud droplet effective radius (CER) and PM2.5. Large-scale measurements of cloud condensation nuclei (CCN) are difficult to obtain on a routine basis, whereas aerosol optical quantities are more readily available (Liu and Li, 2014). However, AOD is not available under cloudy conditions, and AOD cannot represent the aerosol concentrations at the bottom of the cloud, leading to uncertainties in aerosol-cloud-precipitation interaction studies (Liu et al., 2020). Thus, hourly in-situ measurements, such as PM2.5, are an alternative choice to estimate aerosol loading under cloudy conditions.

Aerosol swelling in high humidity cloudy environments (Clarke et al., 2002) is a possible reason behind the large uncertainties in aerosol-cloud interaction (ACI) studies using satellite retrievals (Liu et al., 2018). To address the near-cloud aerosol swelling in the text, we now reference the analysis methodology of Rosenfeld (2000), and have replotted the mean and one standard deviation of CER at each CTT bin in Fig. 8 as below. We defined the clean/polluted days by using surface PM2.5 data, and then displayed the CTT-CER relationship and the occurrence frequency (%) of the CTT in CWP group 9 on clean and polluted days. This avoids the effect of near-cloud aerosol swelling, because PM2.5 observations were at the surface. Figure 8 showed that CTT between 285 and 288 K exhibited a higher occurrence frequency during polluted days, whereas clean days had a higher frequency of CTT between 282 and 285 K. These results suggest that abundant aerosols activated higher concentrations of CCN near surface, thus forming more low-level clouds with smaller cloud droplet size.

In the revised manuscript, we are able provide insights to our research questions, but there are still many uncertainties. For example,  $PM_{2.5}$  is not equal to CCN, satellites cannot observe particle size distribution, and it is difficult ensure our representative aerosols concentrations are present in the cloud. We appreciate that the reviewer suggested such a helpful addition; analysis of rain droplet size distribution provided us another independent verification, which made us more confident in our results.

---

## Author Response (AR2)

We appreciate the suggestion from the referee. Reviewer report are marked as black, our response as blue and changes to the revised manuscript as red.

I am happy to see the results the authors added regarding size-resolved picture of aerosol impacts on precipitation obtained from disdrometer measurements (Figs. 10 and 11). To better illustrate the results, I would suggest to show the droplet size in millimeter, instead of nX, as the horizontal axis label in Figs. 10 and 11.

We appreciate the reviewer for recognizing the value of this work. Many thanks for this suggestion. We re-plotted the droplet size in millimeter instead of nX, as the horizontal axis label in Fig. 10 and Fig. 11. The paragraph has been rewritten as below (lines: 261-266 in the revised manuscript):

Higher populations of raindrops were observed from 0.359 to 0.656 mm (bins n1-n4), with the peak in 0.455 mm (bin n2) for both clean and polluted days (Fig. 10b). The difference is plotted in Fig. 10c. The results illustrate (Fig. 10c) that during polluted days, the droplet numbers appear lower for the smaller raindrop size (< 1.5 mm) compared to clean days and higher for the larger raindrop size (≥ 1.5 mm). A significant reduction in droplet number (decreased from 68 min$^{-1}$ on clean days to 56 min$^{-1}$ on polluted days) was observed in the 0.455 mm (bin n2), corresponding to a reduction in drizzle.

[Figure]

Figure 10: Multiyear (2005-2017) (a) JWD sample number of days in each raindrop size bin, (b) mean droplet number per minute for clean and polluted days, and (c) the differences in the mean droplet number between polluted and clean days. The droplet size for each bin is, in order, 0.359,

0.455, 0.551, 0.656, 0.771, 0.913, 1.116, 1.331, 1.506, 1.665, 1.912, 2.259, 2.584, 2.869, and 3.198 mm.

Lines: 275-277 and lines: 280-284 (in the revised manuscript) have been rewritten as below:

The results are similar with Fig. 10c; the droplet numbers, on polluted days compared to clean days, appear lower for the smaller raindrop size ($\leq 0.771$ mm, bin n5) and higher for the larger raindrop size ($> 0.771$ mm) (Fig. 11a).

We determined that when rainfall was $\leq 1$ mm h$^{-1}$, polluted days accounted for a more significant proportion when raindrop size $\leq 0.5$ mm than clean days (especially in the raindrop size distribution n1, which accounted for 2.3 %) (Fig. 11b). On the other hand, a decreased proportion when raindrop size $> 0.5$ mm was observed during polluted days, as compared with clean days.

[Figure]

Figure 11: Multiyear (2005-2017) differences between polluted and clean days as percentages of the cumulative droplet number distribution for (a) all data and (b) the data with precipitation less than or equal to 1 mm h$^{-1}$. The droplet size bin information of x-axis is same as Fig. 10.